# Waste not, Want not: All-Alive Pruning for Extremely Sparse Networks

## Abstract

Network pruning has been widely adopted for reducing computational cost and memory consumption in low-resource devices. Recent studies show that saliency-based pruning can achieve high compression ratios (*e.g.*, 80–90% of the parameters in original networks are removed) without sacrificing much accuracy loss. Nevertheless, finding the well-trainable networks with sparse parameters (*e.g.*, < 10% of the parameters remaining) is still challenging to network pruning, commonly believed to lack model capacity. In this work, we revisit the procedure of existing pruning methods and observe that *dead connections*, which do not contribute to model capacity, appear regardless of pruning methods. To this end, we propose a novel pruning method, called *all-alive pruning* (*AAP*), producing the pruned networks with only trainable weights. Notably, AAP is broadly applicable to various saliency-based pruning methods and model architectures. We demonstrate that AAP equipped with existing pruning methods (*i.e.*, iterative pruning, one-shot pruning, and dynamic pruning) consistently improves the accuracy of original methods at $128\times$–$4096\times$ compression ratios on three benchmark datasets.

## 1 Introduction

The state-of-the-art neural networks have shown remarkable performance gains on various downstream tasks such as computer vision, natural language processing, and speech recognition. Because neural networks are typically overparameterized, they require high computational cost and memory consumption. Such a nature inherently hinders the deployment of models with excessive parameters on low-end devices such as mobile, embedded, and on-device systems.

Network pruning (Reed, 1993) is the prevalent technique to compress high-capacity models by removing unnecessary units such as weights/filters while maintaining the performance with minimal accuracy loss. Existing pruning methods can be divided into two categories. 1) The first group enforces the sparsity as model regularization (Chauvin, 1988; Weigend et al., 1990; Ishikawa, 1996; Molchanov et al., 2017a; Carreira-Perpiñán & Idelbayev, 2018; Louizos et al., 2018). It is theoretically well-investigated and does not require network retraining. 2) Another group develops saliency criteria to prune less important units (Mozer & Smolensky, 1988; LeCun et al., 1989; Karnin, 1990; Hassibi et al., 1993; Han et al., 2015; Guo et al., 2016; Lee et al., 2019; Park et al., 2020; Evci et al., 2019).

Because of its simple operation and outstanding pruning performance, magnitude pruning (MP) (Han et al., 2015; Narang et al., 2017; Zhu & Gupta, 2018) is the most popular saliency-based pruning method. Recently, the effectiveness of MP is highlighted by the success of the lottery ticket hypothesis (Frankle & Carbin, 2019) and learning rate rewinding (Renda et al., 2020), which achieves less than 1% accuracy loss even after pruning 90% of the parameters. However, all pruning methods show that the trade-off between sparsity and accuracy is significantly degraded, especially at the extremely high sparsity (Gale et al., 2019; Liu et al., 2019; Blalock et al., 2020). Considering the explosive increase in the size of state-of-the-art models for downstream tasks (*e.g.*, GPT-3 (Brown et al., 2020) for machine translation and FixEfficientNet-L2 (Touvron et al., 2020) for image classification), the effective pruning methods at extreme compression ratios must be accomplished, and it is particularly crucial for adopting high-capacity models to low-resource devices (*e.g.* One Laptop per Child).

To break the performance bottleneck at high compression ratios of the state-of-the-art pruning methods, we investigate the procedure of existing pruning methods. Surprisingly, it is revealed that all existing studies overlook the existence of *dead neurons* after pruning – dead neurons are the nodes

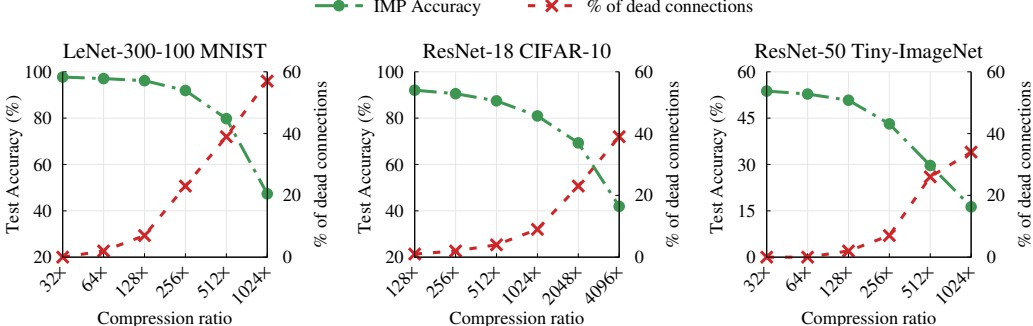

Figure 1: Model accuracy and percentage of dead connections under varying compression ratios: LeNet-300-100 on MNIST, ResNet-18 on CIFAR-10, and ResNet-50 on Tiny-ImageNet. We used IMP with learning rate rewinding (Renda et al., 2020) as the base pruning method. The percentage of dead connections is calculated by the number of dead connections divided by the total number of connections in a given network.

with no input connections or output connections. That is, dead neurons make all their connected weights useless, called *dead connections*. Although Han et al. (2015) have already raised the potential issue of dead neurons, they anticipated that the dead neurons could be negligible after multiple iterative pruning. Unfortunately, we find that this problem is not as simple as their expectation.

Figure 1 depicts prediction accuracy (green line) and the percentage of dead connections (red line) over various compression ratios. For this empirical study, iterative magnitude pruning (IMP) with learning rate rewinding (Renda et al., 2020) is employed as the baseline pruning method. Then, we analyze the correlation between pruning accuracy and the occurrence of dead neurons on three benchmark datasets. We discover two meaningful findings: 1) The dead neurons cannot be successfully removed by iterative pruning, especially at high compression ratios. 2) The severe performance degradation at high compression ratios is strongly correlated with the number of dead connections, *i.e.*, they are inversely proportional to each other.

Based on these valuable findings, we propose *all-alive pruning* (*AAP*), which improves the pruning performance by effectively eliminating dead connections. Specifically, we search dead neurons at the pruning stage by inspecting their gradient flows – when zero gradients are passing through the node, we regard them as dead neurons. Once identified, dead neurons and corresponding dead connections (any weights linked from or to dead neurons) are removed together during the pruning procedure. Note that detecting gradient-based dead neurons can be applied for complex model architectures with skip connections. If we remove more weights than a sparsity threshold, we can revive some weights with the highest saliency scores. As a result, AAP constitutes *all-alive subnetwork*, *i.e.*, all connections in the subnetwork are kept as trainable weights.

To summarize, the key advantage of AAP is two-fold: 1) AAP is versatile – it is broadly applicable to various saliency-based pruning methods and model architectures by minimizing the loss of prediction accuracy. 2) AAP consistently improves the accuracy of the original pruning methods at high compression ratios and breaks the state-of-the-art performance on three benchmark datasets (*i.e.*, MNIST, CIFAR-10, and Tiny-ImageNet).

## 2    RELATED WORK

Network pruning utilizes the highly overparameterized nature of modern neural networks to compress the model. Also, it provides meaningful insight into what leads to the success of neural networks; small subnetworks in the original network can achieve comparable performance and improve generalization effects (Arora et al., 2018).

**Recent advances**. In general, network pruning can be categorized into two groups. The first approach employs the loss function with regularization terms to enforce the sparsity (Chauvin, 1988; Weigend et al., 1990; Ishikawa, 1996). Also, Molchanov et al. (2017a) proposed variational dropout to produce

highly sparse networks and Louizos et al. (2018) reparameterized network weights as the product of weight and a stochastic gate variable. Carreira-Perpiñán & Idelbayev (2018) attempted to minimize the loss using the stochastic version of projected gradient descent.

The second approach utilizes the sensitivity of the loss for weights (LeCun et al., 1989; Karnin, 1990; Hassibi et al., 1993). For saliency-based criteria, Han et al. (2015) proposed iterative pruning based on the magnitude of weights. Despite its simplicity, magnitude pruning shows significant performance gain at high compression ratios (Guo et al., 2016; Narang et al., 2017; Zhu & Gupta, 2018). Recently, Park et al. (2020) proposed lookahead pruning (LAP) that considers the neighbors of connections. While iterative pruning requires pre-trained networks, SNIP (Lee et al., 2019) and GraSP (Wang et al., 2020) proposed to prune the network at initialization. Also, dynamic pruning methods (*e.g.*, SET (Mocanu et al., 2018), DSR (Mostafa & Wang, 2019), and RigL (Evci et al., 2019)) adjust the sparsity level with prune-and-grow cycles. Comprehensive analysis and experimental results for network pruning can be found (Gale et al., 2019; Liu et al., 2019; Blalock et al., 2020).

**Detecting dead connections in network pruning**. Pruning neurons/weights may result in unwanted side-effects, producing useless connections in the pruned networks. In structured pruning, Li et al. (2017) and Liu et al. (2020) attempted to eliminate dead neurons by merely considering the connections between adjacent filters. However, it is not applicable for complex networks with shortcut connections. In unstructured pruning, Han et al. (2015) discussed the existence of dead weights but neglected them. Evci et al. (2019) proposed dynamic pruning, which iteratively updates the weights to identify more useful weights under given computing constraints. Although dynamic pruning eliminates the connections with relatively low importance, it still cannot correctly handle dead connections in each pruning stage. In contrast, this paper devises a new pruning method that removes dead connections and keeps all trainable connections in general networks, including shortcut connections and batch normalization.

## 3 PROPOSED METHOD: ALL-ALIVE PRUNING (AAP)

Given a dataset $\mathcal{D} = \{x^{(i)}, y^{(i)}\}_{i=1}^n$ and a sparsity threshold $\tau$ (*i.e.,* the number of non-zero weights), unstructured pruning can be formulated as the constrained optimization problem.

$$\min_{\mathbf{c}, \mathbf{w}} \mathcal{L}(\mathbf{c} \odot \mathbf{w}; \mathcal{D}) = \min_{\mathbf{c}, \mathbf{w}} \frac{1}{n} \sum_{i=1}^n \ell(\mathbf{c} \odot \mathbf{w}; \mathcal{D}), \quad \texttt{s.t.} \ \ \mathbf{c} \in \{0, 1\}^m \text{ and } ||\mathbf{c}||_0 \le \tau, \quad (1)$$

where $m$ is the number of parameters, and $n$ is the number of samples in $\mathcal{D}$. Also, $\ell(\cdot)$ is the loss function, $\odot$ is the Hadamard product, and $|| \cdot ||_0$ is the $L_0$ norm. $\mathbf{c} \in \{0, 1\}^m$ is the indicator variable to determine whether or not $\mathbf{w}$ is pruned. In other words, $c_j$ indicates whether $w_j$ is active ($c_j = 1$) or inactive ($c_j = 0$).

In this work, we aim at improving saliency-based pruning, where the importance of weights in the network is measured by a pre-defined saliency criterion. Popular criteria include the magnitude of weights (Han et al., 2015), the sensitivity of weights (Lee et al., 2019), and the neighboring connections of weights (Park et al., 2020). Given a saliency criterion, we calculate the importance of weight $w_j$ as a saliency score $s_j$; the higher the saliency score, the more important the weight is. Then, each element of $\mathbf{c}$ is expressed as $c_j = 1$ if $s_j \in \mathcal{S}$ and $c_j = 0$ otherwise, where $j \in \{1, \cdots, m\}$ and $\mathcal{S}$ is a set of $\tau$-largest scores associated with $\mathbf{w}$. That is, $\tau$ connections are kept in the original network, and the rest of the connections are removed from the network.

### 3.1 LIMITATIONS OF EXISTING SALIENCY-BASED PRUNING

Depending on training efficiency, prior studies in saliency-based pruning utilize either *one-shot pruning* or *iterative pruning*. One-shot pruning solely erases selected connections at once and retrains the network. Meanwhile, iterative pruning alternates pruning and retraining cycles by gradually increasing the sparsity levels at multiple steps. Based on this process, iterative pruning can achieve higher accuracy than one-shot pruning for highly compressed networks.

However, pruning weights can lead to unexpected changes in network architecture, and it affects the importance of weights. Although the pruned network can constitute the most useful subnetwork in

terms of selected criteria, the subnetwork may include useless connections, called *dead connections*, as a result of pruning.

To describe the emergence of dead connections, we consider a simple fully-connected network. Here, $w_{ij}^{(l)}$ is a weight connecting two nodes $a_i^{(l-1)}$ and $a_j^{(l)}$ between the $(l-1)$-th layer and the $(l)$-th layer. Based on a backpropagation rule, the gradient of $w_{ij}^{(l)}$ is rewritten as follows:

$$\frac{\partial \mathcal{L}(\mathbf{w}; \mathcal{D})}{\partial w_{ij}^{(l)}} = \frac{\partial a_j^{(l)}}{\partial w_{ij}^{(l)}} \frac{\partial \mathcal{L}(\mathbf{w}; \mathcal{D})}{\partial a_j^{(l)}} = a_i^{(l-1)} \delta_j^{(l)} \quad \text{and} \ \ a_j^{(l)} = \psi \left( \sum_{i=1}^{d} w_{ij}^{(l)} a_i^{(l-1)} + b_j^{(l)} \right), \quad (2)$$

where $\delta_j^{(l)} = \partial \mathcal{L}(\mathbf{w}; \mathcal{D}) / \partial a_j^{(l)}$, and $\psi(\cdot)$ is the activation function. Also, $a_j^{(l)}$ and $b_j^{(l)}$ are an activation and a bias of the $j$-th node at the $(l)$-th layer.

After pruning, the updated network can possibly have a *dead neuron* which appears when the neuron has zero input connections or zero output connections, *i.e.*, either $a_i^{(l-1)}$ or $\delta_j^{(l)}$ is zero:

1) **No input connections**: When $a_i^{(l-1)} = 0$, $a_i^{(l-1)}$ is apparently a dead neuron. According to Eq. (2), the gradient of all output weights $w_{i*}^{(l)}$ are also zeros, and thus can be marked as *dead connections*. Note that, when $a_i^{(l-1)} = b_i^{l-1}$, $a_i^{(l-1)}$ is also regarded as a dead neuron since we can reestablish the network by transferring the bias $b_i^{(l-1)}$ of $a_i^{(l-1)}$ to the bias $b_j^{(l)}$ of $a_j^{(l)}$.

2) **No output connections**: When $\delta_j^{(l)} = 0$, all input weights $w_{*j}^{(l)}$ connected to $a_j^{(l)}$ and the biases $b_j^l$ can be removed.

When the dead neurons are detected, we can also eliminate all useless connections, including the biases and the weights for batch normalization, associated with dead neurons. Note that the dead neurons in the fully-connected layer can be extended as filters in the convolutional networks.

## 3.2 Building All-Alive Subnetwork

The key idea of our pruning method, namely *all-alive pruning* (AAP), is to build *all-alive subnetwork* during the pruning procedure. Toward this goal, we first sort all weights by a saliency-based criterion. We highlight that our pruning method can be applied to various saliency-based criteria. Once the weights with $\tau$-highest scores are determined, we check whether dead connections appear as a side effect of removing the connections. That is, additional dead connections lead to overly pruned networks. To meet the sparsity threshold $\tau$, we can revive some connections with the highest saliency scores among previously pruned connections and update $\mathbf{c}$.

Specifically, the overall process of AAP consists of the following three phases. (In Appendix, Algorithm 2 is the pseudo-code of detecting dead connections in the pruning process.)

1) **Eliminating dead connections**: Choose a set of high-impacted $\tau$ weights according to a saliency-based criterion, and eliminate the rest weights as well as dead connections that are associated with pruned weights.

2) **Reviving new connections**: If the more number of weights are eliminated than a sparsity threshold, we restore some connections with the highest saliency scores, where the restored connections have their original weight values.

3) **Repeat** steps 1–2 until the pruned subnetwork is converged to the *all-alive subnetwork*.

Figure 2 depicts AAP equipped with magnitude pruning (MP). When we prune four weights from the original network, we eliminate the least important connections (dashed lines). While existing pruning methods include dead connections, we eliminate all the incoming (or outgoing) weights of a dead neuron as dead connections (red color) and then revive alternative connections (green color). Finally, all remaining weights are active, thereby the pruned network being all-alive. Please refer to Algorithm 2 for the detailed procedure of AAP in the Appendix.

We observe that AAP has two key advantages. Firstly, all weights in the subnetwork are kept alive without dead connections. That is, we utilize all connections in the subnetwork. Owing to this

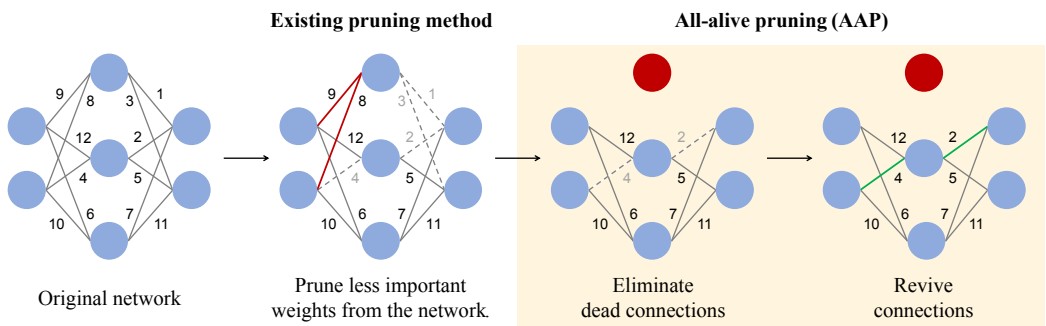

Figure 2: Overall procedure of all-alive pruning (AAP) with a magnitude-based criterion. While existing pruning has a dead neuron (red circle) and its dead connections (red line), AAP safely eliminates dead connections and revive other connections (green line).

compact pruning, we can further compress networks; this is equivalent to improve the prediction accuracy under fixed compression ratios, particularly effective for pruning the networks at extremely high compression ratios. Secondly, since AAP eliminates all the weights associated with dead neurons, AAP results in the memory layout similar to those of structured pruning methods. As a positive byproduct of AAP, it is more compatible with modern memory architectures.

## 4 EXPERIMENTAL RESULTS

We evaluate AAP with a variety of network architectures on MNIST, CIFAR-10, and Tiny-ImageNet datasets. Concretely, we adopt LeNet-300-100 (LeCun et al., 1998), ResNet (He et al., 2016), MobileNetV2 (Sandler et al., 2018), and EfficientNet (Tan & Le, 2019). We employ iterative magnitude pruning (IMP) with learning rate rewinding (Renda et al., 2020), a state-of-the-art pruning method, as our base scheme of AAP. Also, we incorporate AAP into other saliency-based pruning methods such as SNIP (Lee et al., 2019), lookahead pruning (LAP) (Park et al., 2020), and dynamic pruning inspired by RigL (Evci et al., 2019).

**Implementation details**. We use the same environmental setting for all experiments as proposed by Blalock et al. (2020), including initialization values, hyper-parameters, and data augmentation settings. We use Adam optimizer, the batch size of 60, and the constant learning rate of 0.0003 with 50 training epochs for MNIST dataset. For CIFAR-10 dataset, we use the batch size of 128, and the learning rate is initially set to 0.1 and is decayed by 0.1 at every 30 epochs. For Tiny-ImageNet dataset, we use the batch size of 1024, and the learning rate linearly warmed up to its maximum value 0.4 at epoch 5 and decayed by 0.1 at epochs 30, 60, and 80. We use SGD with a momentum of 0.9 and a weight decay rate of 0.0001 for both datasets and train them for 120 epochs and 90 epochs, respectively. For all experiments, we use 10% of the training set as the validation set.

For IMP, we rewind the learning rate schedule to the beginning of the training and retrain from the final values of the weights. For iterative pruning, we prune 50% of the remaining parameters in each step. Exceptionally, since we use the constant learning rate for MNIST dataset, we use *weight rewinding* proposed by Frankle & Carbin (2019). For MNIST dataset, we re-initialize the pruned model's weights to the value of the original model's initial weights.

**Compression ratio**. To measure the degree of network pruning, we adopt a *compression ratio*, which is calculated by *original model size / compressed size*. For instance, when only 1% of the parameters remaining in the compressed network, the compression ratio is 100×. Note that we consider the number of all parameters, including the weights for batch normalization and biases.

### 4.1 EFFECT OF ALL-ALIVE PRUNING

We employ IMP with *learning rate rewinding* (Renda et al., 2020) and *weight rewinding* (Frankle & Carbin, 2019) as the baseline pruning method due to their state-of-the-art performances. At each pruning iteration, we apply AAP after pruning the connections with the lowest magnitude scores. To

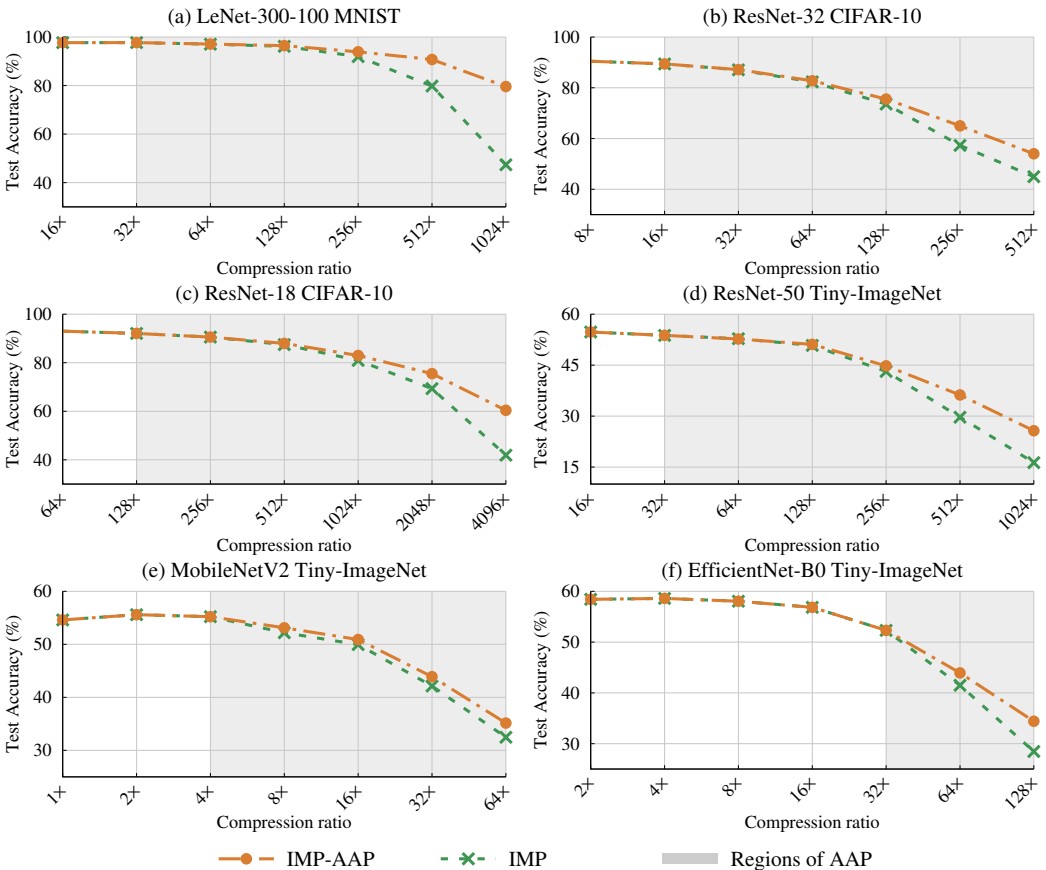

Figure 3: Accuracy comparison between IMP and IMP-AAP across various model architectures and compression ratios by iterative pruning. We use weight rewinding for MNIST dataset and learning rate rewinding for the rest.

validate AAP can alleviate performance degradation at high compression ratios, we start to apply AAP to the point (gray colors in Figure 3), where the performance of pruned networks begins to be worse than the original model. On the basis of IMP, our pruning method, namely IMP-AAP, is compared with IMP in terms of the prediction accuracy under the same compression ratio to evaluate the effectiveness of AAP.

**LeNet-300-100 on MNIST**. LeNet-300-100 consists of the two-layer fully connected (FC) networks with 300 and 100 nodes. We compute the test accuracies of IMP and IMP-AAP at various compression ratios, starting from $16\times$ to $1024\times$. Figure 3(a) clearly showcases the performance gains by adopting AAP, particularly noticeable as the compression ratio exceeds $64\times$. Specifically, IMP-AAP achieves gains of 10.92% at $512\times$ and 32.25% at $1024\times$. From this result, we find two positive effects of AAP: 1) The impressive performance gains at the high compression ratios show the effectiveness of AAP. 2) The success of LeNet-300-100 represents that AAP is effective in fully connected layers.

**ResNet-32 and ResNet-18 on CIFAR-10**. We evaluate AAP on ResNet-32, and ResNet-18 for CIFAR-10 dataset. Figure 3(b) compares the test accuracies of IMP and IMP-AAP over the compression ratios from $8\times$ to $512\times$ on ResNet-32. Despite the rapid drop of IMP in performance after $128\times$, IMP-AAP slows down the degradation of IMP and always achieves higher accuracies than IMP under the same compression ratios. Figure 3(c) depicts the experimental results from ResNet-18. Interestingly, we observe the significant benefits by introducing AAP – IMP-AAP improves IMP by 6% at an extremely high compression ratio $2048\times$. The performance gaps are consistently increasing at higher compression ratios (*e.g.*, greater than $512\times$). From both ResNet-32 and ResNet-18

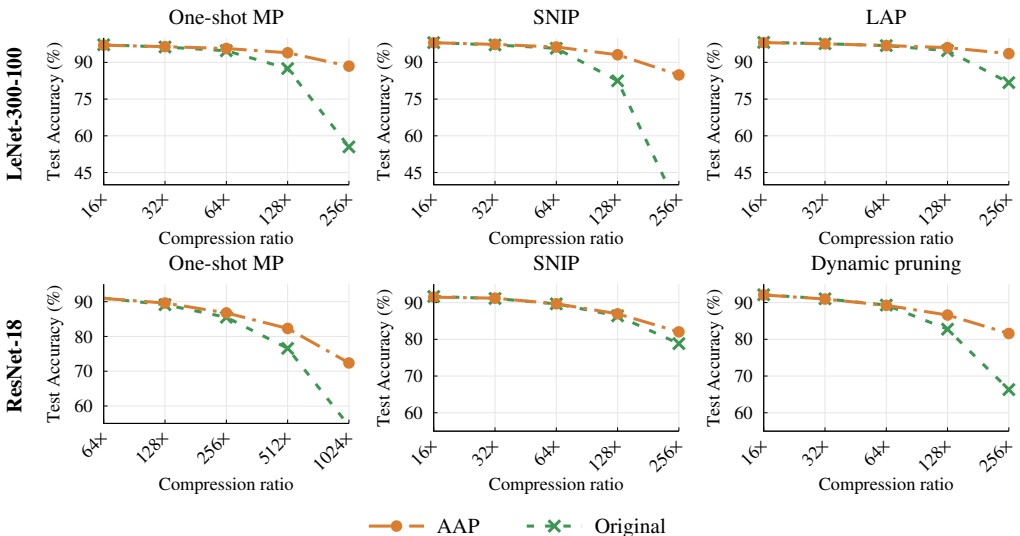

Figure 4: Accuracy of applying AAP with various saliency-based pruning (*i.e.*, one-shot magnitude pruning (One-shot MP) (Frankle & Carbin, 2019; Renda et al., 2020), SNIP (Lee et al., 2019), lookahead pruning (LAP) (Park et al., 2020), and dynamic pruning by RigL (Evci et al., 2019).

experiments, it is consistently observed that the pruned network by IMP-AAP often discards the entire ResBlock except for the skip connection. We conjecture that this is possible because the skip connection allows using the outputs from the previous layers while achieving a high compression ratio. Overall, we confirm that AAP is also effective in compressing convolution layers and the complex structure even with skip connections.

**ResNet-50, MobileNetV2, and EfficientNet-B0 on Tiny-ImageNet**. For Tiny-ImageNet dataset, we employ ResNet-50, MobileNetV2, and EfficientNet-B0 as backbone networks. For ResNet-50, we observe that IMP-AAP improves the trade-off between sparsity and accuracy. When the compression ratio is $512\times$, IMP experiences a severe performance drop, while IMP-AAP improves IMP by $7\%$. In Figures 3(e) and 3(f), we evaluate AAP on MobileNetV2 and EfficientNet-B0 to validate whether or not AAP is still effective in the compact backbone models (*i.e.* having fewer parameters). Due to the highly efficient nature of MobileNetV2 and EfficientNet-B0, both IMP and IMP-AAP suffer from performance degradation when pruning at high compression ratios (*e.g.*, greater than $16\times$). However, IMP-AAP still consistently achieves a meaningful improvement on top of IMP. Concretely, IMP-AAP consistently enjoys the performance improvements about 1-2% at the compression ratios greater than $8\times$ for MobileNetV2. For EfficientNet-B0, the gain by IMP-AAP is also noticeable – IMP-AAP shows 2% improvement on test accuracy at $64\times$ and 6% at $128\times$.

## 4.2 Applicability on Various Saliency-based Pruning

To evaluate the versatility of AAP, we apply AAP on top of existing pruning methods and see whether or not AAP can enjoy the performance benefits of others. We re-implement other saliency-based pruning, referring to the PyTorch implementations, *i.e.*, SNIP[1] and LAP[2]. We apply the same criterion as the weights to the biases for SNIP and use global pruning for LAP. We also validate AAP in dynamic pruning setting, where the prune-and-grow strategy was introduced by RigL (Evci et al., 2019). Specifically, we prune $k$ connections with the lowest magnitude weights then grow $k$ connections with the highest magnitude gradients for every 1500 steps until $3/4$ of the full training epochs. We decay $k$ using a cosine update schedule, where the initial $k$ is 30% of all connections in the network. Then, we apply AAP after each prune-and-grow cycle. Originally, RigL introduced the sparsity strategy for practical training, which assigns the target sparsity level at each layer for preserving low FLOPS. In this experiment, we prune and grow the weights globally, including biases

---

[1] https://github.com/mil-ad/snip
[2] https://github.com/alinlab/lookahead_pruning

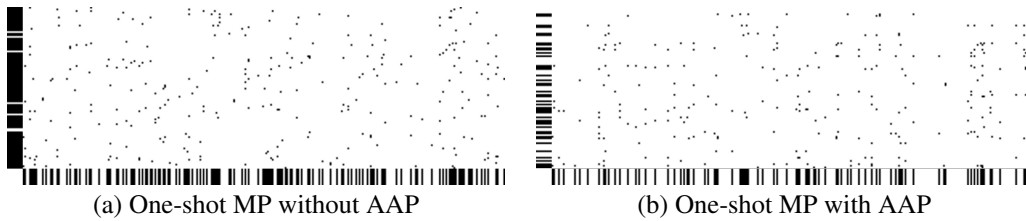

| (a) One-shot MP without AAP | (b) One-shot MP with AAP |

Figure 5: Visualization of remaining weights after applying AAP to one-shot MP at $128\times$ compression ratio using the same unpruned network. The second fully-connected layer ($300 \times 100$) of LeNet-300-100 on MNIST is chosen for the illustration. Each dot denotes the unpruned weight. Whereas 138 out of 300 neurons and 93 out of 100 neurons have unpruned connections in one-shot MP, only 85 out of 300 neurons and 44 out of 100 neurons have unpruned connections after applying AAP. Each achieves 73.23% and 93.12% accuracy, respectively.

and weights for batch normalization unlike RigL. We employ LeNet-300-100 for MNIST dataset and ResNet-18 for CIFAR-10 dataset as backbone networks.

For LeNet-300-100, we conduct comparisons on the basis of one-shot magnitude pruning (One-shot MP), SNIP, and lookahead pruning (LAP). For ResNet-18, the experiment of LAP is not conducted since our method considers all the parameters in the network, including those in skip connections and batch normalization; LAP does not fully consider skip connections and parameters associated with batch normalization. Instead, for ResNet-18, we choose dynamic pruning as a base pruning scheme and apply AAP at each prune-and-grow cycle. For one-shot MP, we apply weight rewinding on LeNet-300-100 and learning rate rewinding on ResNet-18.

Figure 4 depicts the efficiency of AAP over various saliency-based pruning. As expected, since one-shot pruning involves a single pruning stage, any one-shot pruning methods and those with AAP are less effective than IMP. Nevertheless, applying AAP clearly improves various saliency-based, one-shot pruning methods. For one-shot MP on LeNet-300-100 and ResNet-18, we achieve 10% of the accuracy improvement at $128\times$ (from 82.47% to 93.11%) and 6% of the accuracy improvement at $512\times$ (from 76.6% to 82.3%), respectively. Like one-shot MP, AAP incorporates with SNIP well for both LeNet-300-100 and ResNet-18. At $256\times$, AAP outperforms the original SNIP by 33% on LeNet-300-100 (from 55.48% to 88.47%) and 3% on ResNet-18 (from 75.85% to 82.03%). On LeNet-300-100, LAP seems to preserve the good performance even in the high compression ratio; nonetheless, AAP can improve the performance by 12% (81.73% to 93.58%) at $256\times$. Unlike SNIP or LAP, prune-and-grow policy in dynamic pruning leads to architectural changes in the network model. AAP is also successful in improving dynamic pruning; AAP achieves 15% improvements at $256\times$ for dynamic pruning on ResNet-18. From these results, we confirm the efficacy of AAP independently of saliency-based pruning even with dynamic pruning.

In addition to the accuracy gain, Figure 5 compares the weight distribution of the pruned network between one-shot MP and the same one with AAP. While all active neurons from one-shot MP are spread arbitrarily, those from AAP tend to form clusters for more columns and rows. This is notable because AAP does not require explicit or implicit constraints to cluster the weights. We observe that AAP produces a small-dense subnetwork similar to structured pruning (Li et al., 2017; Molchanov et al., 2017b; Liu et al., 2020), which is more favorable for modern memory and computing architectures even with unstructured pruning.

## 5 CONCLUSION

While existing studies mainly focus on developing new saliency-based criteria or optimizing magnitude pruning methods, existing studies do not properly address the dead connections during the pruning process. In this paper, we propose a simple-yet-effective and versatile unstructured pruning method, namely *all-alive pruning* (*AAP*), to eliminate dead connections and make all weights in the subnetwork trainable. In experimental results, AAP consistently improves various saliency-based pruning methods with different model architectures at $128\times$–$4096\times$ compression ratios, achieving state-of-the-art performance on several benchmark datasets.

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

## A    DETAILED ALGORITHMS OF THE PROPOSED METHOD

**Finding dead connections**. Given a binary mask vector $\mathbf{c} \in \{0,1\}^m$ and the corresponding parameter $\mathbf{w} \in \mathbb{R}^m$, we find a dead connection vector $\mathbf{d} \in \{0,1\}^m$, where $d_{ij}^{(l)}$ is 1 if $w_{ij}^{(l)}$ is a dead connection; $d_{ij}^{(l)}$ is 0 otherwise. When the weight matrix is higher than two-dimensional, *e.g.* filters in CNN, the first two dimensions are regarded as input and output dimensions. To simplify the notation, $\mathbf{g}_{i*} = 0$ is same as $\mathbf{g}_{i:*:*:*} = 0$.

Algorithm 1 is the pseudo-code of finding dead connections. First of all, we preprocess a network to check the gradient of weights. In this preprocessing stage, the activation function is regarded as an identity function. Besides, we set all weights to be positive and ignore all bias terms to avoid false positive neurons (*i.e.*, detecting the false dead neurons, although it is not dead neurons). After that, we update the weights by considering the given mask vector and compute the gradient $g(\cdot)$ of $\mathbf{w}$ by taking the all-one vector as input. After the dead connection vector is initialized with a zero vector, we update the gradient for dead connections.

**Pseudo-code of all-alive pruning**. Algorithm 2 is the pseudo-code of the all-alive pruning algorithm. Given a network, we compute the saliency score using an arbitrary saliency criterion, *e.g.* magnitude pruning, SNIP (Lee et al., 2019), and lookahead pruning (LAP) (Park et al., 2020). Then we update $\mathbf{c}$ using the saliency score, *i.e.* the weights with the lowest scores are set to 1. For the updated sub-network, we find the dead connections and recalculate the saliency scores accordingly. That is, if the weight is identified as a dead connection, its saliency score is set to 0. We repeat eliminating dead connections and making *all-alive subnetwork* by given saliency score until all connections are kept alive. Then we prune the network with the finally obtained mask vector.

---

**Algorithm 1** *Finding the dead connections*

---

**Input:** Binary mask $\mathbf{c} \in \{0,1\}^m$, Binary dead connections $\mathbf{d} \in \{0,1\}^m$, model parameters $\mathbf{w} \in \mathbb{R}^m$
**Output:** Updated binary dead connections $\mathbf{d} \in \{0,1\}^m$ corresponding to $\mathbf{w} \in \mathbb{R}^m$

1:  **Preprocess**: Make all weights positive, remove all biases.
2:  $\tilde{\mathbf{w}} \leftarrow \mathbf{w} \odot \mathbf{c}$.                                          ▷ Update the pruned network with $\mathbf{c}$.
3:  Compute the gradient $\mathbf{g}(\tilde{\mathbf{w}}; \mathbb{1})$.                    ▷ Update the gradient $\mathbf{g}$ for all-one input data.
4:  **for** $l = 1$ to $N$ **do**
5:      $\mathbf{d}_{ij}^{(l)} \leftarrow 1$   if   $\mathbf{g}_{i*}^{(l)} = 0$ or $\mathbf{g}_{*j}^{(l)} = 0$.
6:  **end for**

---

---

**Algorithm 2** *All-alive pruning*

---

**Input:** Model parameters $\mathbf{w} \in \mathbb{R}^m$, sparsity threshold $\tau$
**Output:** All-alive subnetwork $\mathbf{w}$

1:  Compute saliency $\mathbf{s}$ for $\mathbf{w}$.                                   ▷ Use an arbitrary saliency criterion.
2:  Initialize binary dead connections $\mathbf{d} \in \{0,1\}^m$ as zero.
3:  **repeat**
4:      $\mathcal{S} \leftarrow$ top-$\tau$ connections by $\mathbf{s}$.
5:      Initialize $\mathbf{c}$ as zero and update $\mathbf{c}$ for $\mathcal{S}$.                  ▷ Set top-$\tau$ connections in $\mathcal{S}$ as $c_j = 1$.
6:      Update dead connections $\mathbf{d}$ for $\mathbf{w}$ and $\mathbf{c}$.                              ▷ Refer to Algorithm 1.
7:      **for** $l = 1$ to $N$ **do**
8:          $\mathbf{s}_{ij}^{(l)} \leftarrow 0$   if   $\mathbf{d}_{ij}^{(l)} = 1$.                ▷ Update the score of dead connections as $s_{ij}^{(l)} = 0$.
9:      **end for**
10: **until** all connections in $\mathbf{c}$ is alive.
11: $\mathbf{w} \leftarrow \mathbf{w} \odot \mathbf{c}$

---

## B    MODEL ARCHITECTURE DETAILS AND HYPERPARAMETER SETTINGS

We modify the input and output layers of model architectures to handle a smaller size of input data. For CIFAR-10, we use a $3 \times 3$ filter with a stride of 1 at the first convolutional layer of ResNet. Because the image resolution of the Tiny-ImageNet dataset is larger than that of the CIFAR-10

dataset, we double the stride of the first convolutional layer. For MobileNetV2 and EfficientNet-B0, we use a stride of 1 instead of 2 at the first convolutional layer. Also, we adjust the number of output nodes for all networks according to the number of labels for each dataset.

We re-implement SNIP and LAP, referring to the PyTorch implementations. We apply the same criterion as the weights to the biases for SNIP. For LAP, we use global pruning (*i.e.* prune the entire network at once) instead of layer-wise pruning for fair comparisons with other pruning methods; the original LAP chooses layer-wise pruning. Whereas SNIP and LAP do not fully consider the biases and the weights for batch normalization, our AAP equally treats and removes all parameters in the network.

| Dataset | Model | #Params | Training settings | Learning rate | | Accuracy |
|---------|-------|---------|-------------------|---------------|---|----------|
| MNIST | LeNet-300-100 | 267K | Adam optimizer
Batch size: 60 | 0.0012 | | $97.76_{\pm 0.19}$ |
| CIFAR-10 | ResNet-32 | 0.46M | SGD optimizer
Batch size: 128
Momentum: 0.9
Weight decay: 0.0001 | $lr = \begin{cases} 0.1 & 0 < t \leq 30 \\ 0.01 & 30 < t \leq 60 \\ 0.001 & 60 < t \leq 90 \\ 0.001 & 90 < t \leq 120 \end{cases}$ | | $90.14_{\pm 0.69}$ |
| | ResNet-18 | 11.2M | | | | $91.98_{\pm 0.06}$ |
| Tiny-ImageNet | ResNet-50 | 23.9M | SGD optimizer
Batch size: 1024
Momentum: 0.9
Weight decay: 0.0001 | $lr = \begin{cases} 0.4 \cdot \frac{t}{5} & 0 < t \leq 5 \\ 0.4 & 5 < t \leq 30 \\ 0.04 & 30 < t \leq 60 \\ 0.004 & 60 < t \leq 80 \\ 0.0004 & 80 < t \leq 90 \end{cases}$ | | $53.55_{\pm 1.66}$ |
| | MobileNetV2 | 2.5M | | | | $54.60_{\pm 0.50}$ |
| | EfficientNet-B0 | 4.3M | | | | $56.23_{\pm 0.47}$ |

Table 1: Hyperparameters and training schedules for each dataset. $t$ indicates the training epochs. All accuracy is for the original model and is the average of the three trials.

## C DETAILED EXPERIMENTAL RESULTS

We report numerical results depicted in Figure 3. Note that all the results are computed by averaging three trials with different random seeds. In all tables, subscripts denote standard deviations, and bracketed numbers indicate the accuracy gains obtained by the proposed model over the original IMP.

| | 32× | 64× | 128× | 256× | 512× | 1024× |
|---|-----|-----|------|------|------|-------|
| % params | 3.13% | 1.56% | 0.78% | 0.39% | 0.20% | 0.10% |
| IMP | $97.72_{\pm 0.08}$ | $97.06_{\pm 0.19}$ | $96.18_{\pm 0.15}$ | $91.94_{\pm 0.79}$ | $79.80_{\pm 2.76}$ | $47.35_{\pm 10.04}$ |
| IMP-AAP | – | $\mathbf{97.11_{\pm 0.21}}$
(+0.05%) | $\mathbf{96.44_{\pm 0.25}}$
(+0.26%) | $\mathbf{93.91_{\pm 0.44}}$
(+1.97%) | $\mathbf{90.72_{\pm 0.69}}$
(+10.92%) | $\mathbf{79.60_{\pm 2.75}}$
(+32.25%) |

Table 2: Accuracy of LeNet-300-100 on MNIST. The initial unpruned network has 267K parameters and 97.76% average accuracy.

| | 16× | 32× | 64× | 128× | 256× | 512× |
|---|-----|-----|-----|------|------|------|
| % params | 6.25% | 3.13% | 1.56% | 0.78% | 0.39% | 0.20% |
| IMP | $89.40_{\pm 0.58}$ | $87.06_{\pm 0.49}$ | $82.33_{\pm 0.70}$ | $73.57_{\pm 0.75}$ | $57.30_{\pm 0.70}$ | $44.96_{\pm 2.16}$ |
| IMP-AAP | – | $\mathbf{87.16_{\pm 0.54}}$
(+0.10%) | $\mathbf{82.76_{\pm 0.58}}$
(+0.43%) | $\mathbf{75.56_{\pm 0.72}}$
(+1.99%) | $\mathbf{64.99_{\pm 1.44}}$
(+7.69%) | $\mathbf{53.99_{\pm 1.75}}$
(+9.03%) |

Table 3: Accuracy of ResNet-32 on CIFAR-10. The initial unpruned network has 0.46M parameters and 90.14% average accuracy.

| | 128× | 256× | 512× | 1024× | 2048× | 4096× |
|---|---|---|---|---|---|---|
| % params | 0.78% | 0.39% | 0.20% | 0.10% | 0.05% | 0.02% |
| IMP | 92.07±0.10 | **90.57**±0.38 | 87.47±0.33 | 80.96±0.09 | 69.32±1.54 | 41.92±11.98 |
| IMP-AAP | — | 90.56±0.28 (-0.01%) | **87.96**±0.35 (+0.49%) | **82.98**±0.24 (+2.02%) | **75.50**±1.13 (+6.18%) | **60.40**±2.12 (+18.48%) |

Table 4: Accuracy of ResNet-18 on CIFAR-10. The initial unpruned network has 11.2 parameters and 91.98% average accuracy.

| | 32× | 64× | 128× | 256× | 512× | 1024× |
|---|---|---|---|---|---|---|
| % params | 3.13% | 1.56% | 0.78% | 0.39% | 0.20% | 0.10% |
| IMP | 53.76±1.31 | **52.79**±0.86 | 50.81±0.75 | 43.15±0.58 | 29.68±0.70 | 16.31±2.29 |
| IMP-AAP | — | 52.70±1.09 (-0.09%) | **51.11**±0.86 (+0.30%) | **44.79**±0.75 (+1.64%) | **36.22**±0.88 (+6.54%) | **25.70**±0.81 (+9.39%) |

Table 5: Top-1 accuracy of ResNet-50 on Tiny-ImageNet. The initial unpruned network has 23.9M parameters and 53.55% average top-1 accuracy.

| | 32× | 64× | 128× | 256× | 512× | 1024× |
|---|---|---|---|---|---|---|
| % params | 3.13% | 1.56% | 0.78% | 0.39% | 0.20% | 0.10% |
| IMP | 77.15±0.98 | **77.44**±0.67 | 76.26±0.73 | 70.61±0.47 | 57.01±0.32 | 38.90±3.19 |
| IMP-AAP | — | 77.10±0.71 (-0.34%) | **76.48**±0.83 (+0.22%) | **71.53**±0.88 (+0.92%) | **63.88**±0.83 (+6.87%) | **52.34**±0.73 (+13.44%) |

Table 6: Top-5 accuracy of ResNet-50 on Tiny-ImageNet. The initial unpruned network has 23.9M parameters and 76.63% average top-5 accuracy.

| | 2× | 4× | 8× | 16× | 32× | 64× |
|---|---|---|---|---|---|---|
| % params | 50.00% | 25.00% | 12.50% | 6.25% | 3.13% | 1.56% |
| IMP | 55.60±0.02 | 55.22±0.61 | 52.19±0.33 | 49.96±0.34 | 42.13±0.78 | 32.47±0.21 |
| IMP-AAP | — | — | **53.12**±0.12 (+0.93%) | **50.90**±0.38 (+0.94%) | **43.90**±0.23 (+1.77%) | **35.15**±1.22 (+2.68%) |

Table 7: Top-1 accuracy of MobileNetV2 on Tiny-ImageNet. The initial unpruned network has 2.5M parameters and 54.60% average top-1 accuracy.

| | 2× | 4× | 8× | 16× | 32× | 64× |
|---|---|---|---|---|---|---|
| % params | 50.00% | 25.00% | 12.50% | 6.25% | 3.13% | 1.56% |
| IMP | 79.49±0.23 | 79.58±0.05 | 77.44±0.21 | 75.70±0.53 | 69.22±0.41 | 60.42±0.36 |
| IMP-AAP | — | — | **77.75**±0.32 (+0.31%) | **76.10**±0.27 (+0.40%) | **70.97**±0.33 (+1.75%) | **63.00**±1.32 (+2.58%) |

Table 8: Top-5 accuracy of MobileNetV2 on Tiny-ImageNet. The initial unpruned network has 2.5M parameters and 78.83% average top-5 accuracy.

## D  SPARSITY PATTERNS IN LENET-300-100

We visualize the change of the pruned weights in LeNet-300-100 over compression ratios (from 64× to 256×). In all figures, each dot denotes the unpruned weight on the $300 \times 100$ matrix. To observe the differences between pruning methods, we perform one-shot MP in the same network. As we expected, all-alive pruning behaves similar to structured pruning by removing *dead neurons*. Note

| | 16× | 32× | 64× | 128× |
|---|---|---|---|---|
| % params | 6.25% | 3.13% | 1.56% | 0.78% |
| IMP | 56.85±0.36 | 52.31±0.33 | 41.49±0.14 | 28.46±0.74 |
| IMP-AAP | – | – | **43.93±0.24** (**+2.44%**) | **34.41±0.26** (**+5.95%**) |

Table 9: Top-1 accuracy of EfficientNet-B0 on Tiny-ImageNet. The initial unpruned network has 4.3M parameters and 56.23% average top-1 accuracy.

| | 16× | 32× | 64× | 128× |
|---|---|---|---|---|
| % params | 6.25% | 3.13% | 1.56% | 0.78% |
| IMP | 80.37±0.14 | 77.69±0.57 | 69.61±0.43 | 55.93±0.82 |
| IMP-AAP | – | – | **70.62±0.35** (**+1.01%**) | **62.06±0.04** (**+6.13%**) |

Table 10: Top-5 accuracy of EfficientNet-B0 on Tiny-ImageNet. The initial unpruned network has 4.3M parameters and 80.24% average top-5 accuracy.

that the structured pruning makes the pruned sub-network to be small-dense. Similarly, AAP also produces the sub-network to have a small-dense structure because the dots are clustered along with either columns or rows – if the black bars along with the vertical or horizontal axis are scarce, the sub-network becomes small-dense. Interestingly, this tendency becomes prominent with an extremely high compression ratio (*e.g.* 256×) as shown in Figure 7. In other words, AAP achieves 89.65% accuracy even with weights connected with 36 out of 300 neurons and 29 out of 100 neurons.

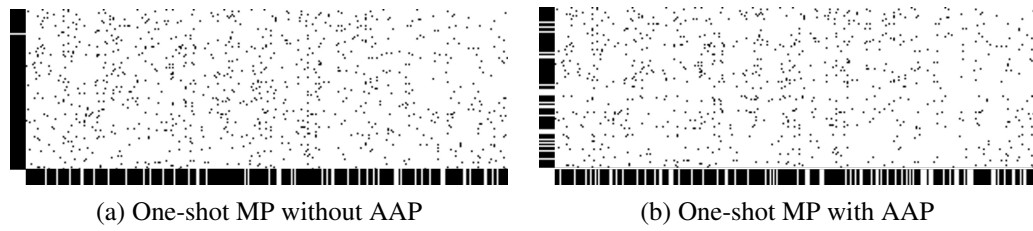

| (a) One-shot MP without AAP | (b) One-shot MP with AAP |
|---|---|

Figure 6: Visualization of remaining weights in the second fully-connected layer of LeNet-300-100 after applying AAP to one-shot MP at 64× compression ratio. 250 out of 300 neurons and 99 out of 100 neurons have unpruned connections in one-shot MP, and 220 out of 300 neurons and 77 out of 100 neurons have unpruned connections after applying AAP. Each achieves 96.01% and 96.48% accuracy after pruning, respectively.

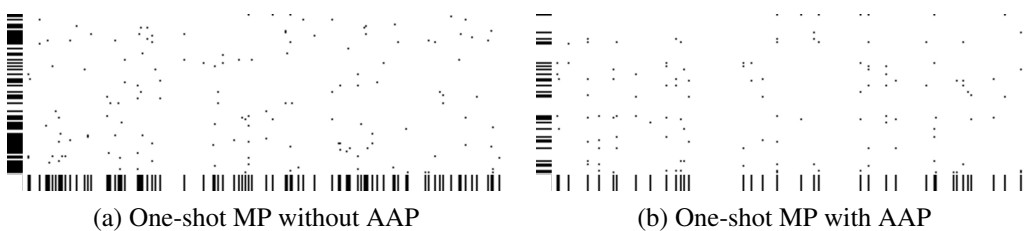

| (a) One-shot MP without AAP | (b) One-shot MP with AAP |
|---|---|

Figure 7: Visualization of remaining weights in the second fully-connected layer of LeNet-300-100 after applying AAP to one-shot MP at 256× compression ratio. 88 out of 300 neurons and 64 out of 100 neurons have unpruned connections in one-shot MP, and 36 out of 300 neurons and 29 out of 100 neurons have unpruned connections after applying AAP. Each achieves 38.06% and 89.65% accuracy after pruning, respectively.

# E    THEORETICAL STUDY OF ALL-ALIVE PRUNING

## E.1    CONVERGENCE OF ALL-ALIVE PRUNING

In this section, we study the convergence of the proposed all-alive pruning. In all iterations of AAP (Algorithm 2), we mask all dead connections found in the previous iteration to avoid reviving the dead connections. That is, the sub-network derived from one iteration of AAP should have a smaller number of connections than the previous iteration, *i.e.*, the number of connections in the sub-network excluding dead connections for $t$ iteration is followed as:

$$0 < n(\theta_t) = n(\theta_{t-1}) - n(D_t^\tau) < n(\theta_{t-1}), \tag{3}$$

where $\theta_t$ and $D_t^\tau$ are the parameters in the sub-network and the found dead connections respectively at $t$ iteration of AAP. Since the number of connections considered for each iteration decreases, AAP can always converge. It is observed that the 2-3 iterations are repeated in most cases.

## E.2    ELIMINATING DEAD CONNECTIONS FROM LOWER COMPRESSION RATIOS

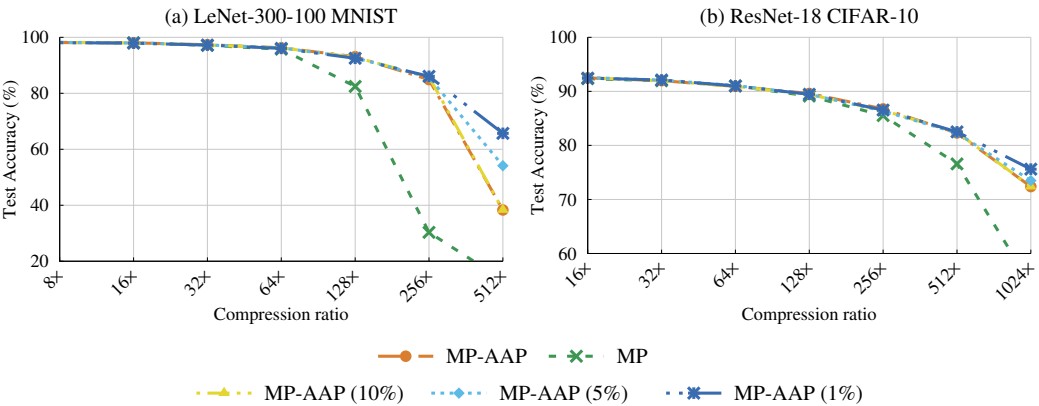

Figure 8: Accuracy of applying AAP incrementally in one-shot magnitude pruning. AAP(1%) denotes that the process of pruning $\frac{n(\theta)-\tau}{100}$ connections and eliminating corresponding dead connections was repeated in order to reach the final sparsity threshold $\tau$. Note that we only eliminate dead connections without any training process differs from the *iterative pruning*.

Without knowing the dependency of all connections, finding the optimal sub-network with the highest saliency scores without any dead connections is hard. All-alive pruning is proposed to obtain a more efficient sub-network with the existing saliency criteria; nevertheless, it also cannot guarantee the sub-network with the highest scores. In this section, we deal with the more efficient sub-network by applying AAP incrementally from lower compression ratios (higher sparsity thresholds). With the given sparsity threshold $\tau$, the total saliency scores of the sub-network obtained by AAP is followed as:

$$S^\tau = \sum TopK(|\theta - D^\tau|, \tau), \tag{4}$$

where $TopK(v, k)$ indicates the top-$k$ elements of set $v$. Note that the saliency scores of the dead connections could be considered as zero as they cannot properly join the regular training. If we eliminate the dead connections for $\tau' > \tau$ sparsity threshold first, then pruning up to $\tau$ again, the dead connections found for the final sub-network is changed as:

$$D^{\tau' \to \tau} = D^{\tau + n(D^{\tau'})} \subseteq D^\tau, \tag{5}$$

since the dead connections found by the previous iteration decrease the removal of the connections of the sub-network needed for satisfying the sparsity threshold, in the same manner with the motivation of AAP. It is obvious that the dead connections $D^{\tau + n(D^{\tau'})}$ caused by fewer pruned connections than

$\tau$ is a subset of $D^\tau$. The total saliency scores of the sub-network $S^{\tau' \to \tau}$ with the corresponding dead connections $D^{\tau' \to \tau}$ are changed to:

$$
\begin{aligned}
S^{\tau' \to \tau} &= \sum TopK(|\theta - D^{\tau' \to \tau}|, \tau) \\
&= \sum TopK(|\theta - D^{\tau + n(D^{\tau'})}|, \tau) \\
&\geq S^\tau = \sum TopK(|\theta - D^\tau|, \tau) \because D^{\tau + n(D^{\tau'})} \subseteq D^\tau.
\end{aligned}
\tag{6}
$$

According to Eq. (6), we can see that the dead connections are eliminated in smaller units, the higher the total saliency scores can be obtained; eliminating dead connections whenever prune each connection can obtain the sub-network with much higher saliency scores. However, this does not mean that we can get an optimal network with the highest saliency scores, *e.g.*, the connection with the lowest saliency score can still be needed to alive, and the existing saliency-based pruning methods do not allow to make it alive.

To validate the effectiveness of saving the saliency scores by applying AAP incrementally from the lower compression ratios (higher sparsity thresholds), we conducted the experiments in one-shot magnitude pruning. We satisfied the sparsity threshold $\tau$ over several pruning steps without any training process, but only with eliminating newly appearing dead connections. Figure 8 depicts the experiments of the AAP with smaller units. For example, AAP(1%) denotes that we repeat the process of **1) prune** $\frac{n(\theta) - \tau}{100}$ **connections, then 2) eliminate the corresponding dead connections** 100 times for satisfying the sparsity threshold $\tau$. Interestingly, we can observe that the sub-network with the higher total saliency scores can achieve better accuracy at high compression ratios. Note that the existing pruning always outputs the same sub-network even with incremental pruning since they eliminate the connections with the lowest saliency scores without considering the effect of pruning on other connections.

