# OpenReview forum: "Waste not, Want not: All-Alive Pruning for Extremely Sparse Networks"
_ICLR.cc/2021/Conference — Reject_

### Official Review · AnonReviewer1 · 2020-10-27
**An effective but trivial idea**

**Rating:** 5
**Confidence:** 4

**Review:**

The paper proposes to remove dead neurons and their connected parameters through a very simple check while reviving pruned (salient) parameters up to the prespecified sparsity level, such that the sparse network obtained could perform better. The main (and perhaps the single major) contribution of this work is in its demonstration that such a simple method is indeed effective for different pruning methods on various network architectures and datasets. The proposed method (AAP) can perhaps be considered as a generic post-processing step that could be equipped to any pruning method leaving dead neurons.

While the result is not unimportant, a few concerns remain. AAP is literally just checking for zero gradients (Line 6 in Algorithm 1) which by itself is very trivial and thus it remains questionable if this is really enough contribution as a sole idea for a conference paper (rather than for a workshop or so to speak); in other words, the idea is too generic that it could be done without having a read  of this paper. Moreover, except for visualization of remaining parameters, the idea is mainly demonstrated only for the performance, and it lacks in-depth analysis, ablations, and insights including for instance, why existing methods produce dead neurons, what’s the cost of AAP (or N value) in practice and theory, etc. So I’m leaning slightly towards rejecting it for limited novelty.

---

> ### Author Response · Authors · 2020-11-23
> **Response to R1**
>
> We sincerely appreciate your comments and efforts in reviewing our paper. Mostly, thanks for the positive comments about clear motivations (R3, R4), the effectiveness of the methods (R1, R2, R3, R4), and the good-written paper(R2, R3, R4). We revised our manuscript and added a study on the convergence of AAP and an iterative approach to obtain a sub-network with higher saliency scores in the Appendix.
>
> We responded to questions/concerns from the reviewer below:
>
> ---
>
> **Q1.** The idea is too generic.
>
> **A1.** While many studies rely on the magnitude pruning, we believe that finding the 'perfect' saliency score for unstructured pruning is essential. Even a large amount of methods for improving the performance such as [1, 2] are proposed, the studies about the limitations of their saliency scores are not adequately studied. Our goal is to address their limitations, and we propose a novel approach to overcome such limitations. Ironically, the existence of a general method like AAP, which improves various pruning methods, indicates that there are still many points to be studied in pruning.
>
> ---
>
> **Q2.** More research related to the dead connections would be needed.
>
> **A2.** We agree that there are more studies to understand the true meaning of dead connections. However, until now, it is difficult to figure out why dead connections occur and how many of them occur in a given sparsity.
> The ultimate goal of pruning is to find the most efficient saliency criteria that do not waste any connections. Towards this goal, we propose a new baseline for various unstructured pruning by showing that the dead connections appear even with the iterative pruning with learning rate rewinding. In the case of the estimated costs, we observe that 2-3 iterations are mostly enough to eliminate all dead connections in the experiment. It means that it only takes about 2-3 times of inference. We also hope that our work will present a novel intuition along with the perception of dead connections in future pruning studies. We would continue to add more analysis results if time permits.
>
> ---
>
> [1] Jonathan Frankle and Michael Carbin. The lottery ticket hypothesis: Finding sparse, trainable neural networks. In International Conference on Learning Representations, 2019.
>
> [2] Alex Renda, Jonathan Frankle, and Michael Carbin. Comparing rewinding and fine-tuning in neural network pruning. In International Conference on Learning Representations, 2020.

---

### Official Review · AnonReviewer2 · 2020-10-28
**A necessary improvement to pruning, but method needs justification**

**Rating:** 5
**Confidence:** 3

**Review:**

== Summary ==

The submission deals with eliminating neurons in a network where either a) all the input connections xor b) all the output connections have been pruned. When this is the case, the unpruned a) output or b) input connections are unused and can also be pruned: and the freed parameter budget used for other more useful connections. This is shown to improve the accuracy of pruned networks at a given sparsity ratio, especially for very high levels of sparsity.

== Strengths ==

**Significance:** Nearly all neural network pruning approaches should consider this kind of improvement. Zombie (dead but un-pruned) connections in a pruned model will incur a cost with no benefit. As observed in the related work, some previous literature does explicitly ensure there are no dead neurons, but this is not universal. The authors propose a fairly general kind of approach, so it has potential for adoption fairly broadly across sparsity methods.

**Experimental results** Overall, the experiments are well-executed and show a good effect. The effect of the proposed method, varying with the model size, on the accuracy is clearly shown. The results should be very reproducible, with explicit random seeding and multiple trials.

**Clarity** The core method is described very straightforwardly and completely. Provided code is also very clear, and runnable out-of-the-box, and helped provide some answers.

== Weaknesses ==

While this is a well-executed investigation of the idea, I'm not convinced that using the gradient to detect dead neurons is the right approach.

**Complexity** It's not a great deal of additional complexity, but it seems unjustified. Fig 2 itself illustrates that the "dead" connections could be found by directly inspecting whether the weights are non-zero. The most thorough explanation, that I can find, as to why the authors did not use the "basic" approach is that "it is not applicable for complex networks with shortcut connections." But (Liu et. al. 2020), *do* consider a ResNet network. That paper also refers also to:

He et. al. "Channel Pruning for Accelerating Very Deep Neural Networks" ICCV 2017

that goes into how shortcuts affect pruning filters in more details.

**Possible drawbacks** It seems possible that an activation or gradient may be zero in one iteration simply due to a filter not being active on examples in a *that minibatch*. So measured, a dead connection on this minibatch does not imply that it will be dead for all possible examples. Especially given that a filter may be preferentially responding to a particular kind of image structure, see for example:

Zieler & Fergus "Visualizing and Understanding Convolutional Networks" ECCV 2014

It has also already been observed that some pruning can essentially discard accuracy on example with rarer properties, to preserve performance on data more similar to the mode, see:

Hooker et. al. "What Do Compressed Deep Neural Networks Forget?"

and it seems possible that the submission's method will tend to work this way to an even greater extent.

This also possibly isn't measured well by accuracy on small-scale experiments such as MNIST and Tiny-ImageNet: with a larger model and larger datasets I'd guess that probability is greater that a dead connection will be spuriously identified somewhere in the model.

I'd especially appreciate any rebuttal that elaborates on why this method was chosen, rather than directly analyzing the elements of the weights $\mathbf{w}$ or mask $\mathbf{c}$ to determine if any node has all-zero input or output connections.

== Misc Comments and Questions ===
(not relevant to score)

  * Grammar in "we experiment with dynamic pruning instead of LAP is not compared since our method considers all the parameters in the network" on page 8?
  * Typo "respecitvely" on p8
  * Why use np.abs in pruning/magnitude.py instead of torch.abs?
  * What is the meaning of plus/minus notation in tables in Appendix C? Standard deviation? Difference between highest and lowest?
  * Is it certain that the all-alive pruning will converge? I.e. is there an obvious reason that the outer loop in Algorithm 2 is known to always terminate in finite time?

== Reason for Rating ==

Eliminating "dead" neurons, and not wasting the unused connections on them, is obviously a good choice when pruning a network. The central novel contribution of the paper, though, is in specifically how this is achieved. The method proposed in not completely justified by the submission.

== After Rebuttal ==

Q1 provides a helpful clarification. Some of the more concrete possible drawbacks listed above, especially those about pruning "false positives," no longer seem as much of a concern.

Some of the earlier investigation that the authors report, on more direct methods for finding dead connections, may need to be given a more complete treatment in the paper itself. I am not convinced that the direct methods cannot be done on arbitrary architectures, given that previous literature has managed to in a wide variety of examples. Without considering the simpler techniques, the leap to a more complex method for what could be a relatively simple task doesn't have the necessary support.

---

> ### Author Response · Authors · 2020-11-23
> **Response to R2**
>
> We sincerely appreciate your comments and efforts in reviewing our paper. Mostly, thanks for the positive comments about clear motivations (R3, R4), the effectiveness of the methods (R1, R2, R3, R4), and the good-written paper(R2, R3, R4). We revised our manuscript and added a study on the convergence of AAP and an iterative approach to obtain a sub-network with higher saliency scores in the Appendix.
>
> We responded to questions/concerns from the reviewer below:
>
> ---
>
> **Q1.** The appearance of the dead connections and 'mini-batch'.
>
> **A1.** We did not use any min-batch from the dataset to inspect the dead connections. Our method is worked independent with the data and always outputs the same sub-network with the given sparsity threshold and the saliency scores of the connections of the network. The reason we did not use the mini-batch was that, as you were concerned, it could produce results that depend on such data.
>
> Concretely, we make all weights positive and give them positive value as input to always make the gradients flow if there are connections to find the zero-input/outputs. Of course, this pre-process is applied when to find dead connections; we use the original weights after we found the all-alive subnetwork structure. Before proceeding with experiments of AAP, we verified that the elimination of found dead connections results in the same output as the original network. For the final version, we will review the paper again to express this clearly.
>
> ---
>
> **Q2.** The reason for inspecting gradient, not weights or the connections.
>
> **A2.** In earlier stages, we used the weights or the connections directly to find dead connections and check the effectiveness of our proposed method. However, we found that it is impractical to apply our method generally to the model with the structure that creates a computation graph different from general MLP and convolutional layers such as shortcut connections, depth-wise convolutional layers. By inspecting the connections with the zero-gradients, we can easily detect the dead connections in various architectures without considering all structures. It also does not depend on some data, as mentioned in Q1.
>
> [1] also tried to find the effects of the filter pruning, but they allow only the structure with a limited arrangement of shortcut connections. (Note that they consider this before structured pruning, while we have to inspect the dead connections after the various unstructured pruning methods.) By inspecting gradients, our method can be network-agnostic and can be easily applied to the more complex even with the depth-wise convolution layer, global average pooling layer. Our method can be applied to the more general architecture other than CNN, such as RNN and Transformer.
>
> ---
>
> We also address some minor issues (grammar error, typo) you mentioned. Appreciate the detailed review. The mixing of np.abs and torch.abs in the code is due to several migrations and modifications, and there is no difference in functionality. Sorry if there was any confusion in reading the code.
>
> ---
>
> [1] Hao Li, Asim Kadav, Igor Durdanovic, Hanan Samet, and Hans Peter Graf. Pruning filters for efficient convnets. In International Conference on Learning Representations, 2017.

---

### Official Review · AnonReviewer3 · 2020-10-28
**Simple yet effective pruning approach focusing on removing dead connections in a network**

**Rating:** 7
**Confidence:** 3

**Review:**

Summary:
This paper talks about a novel network pruning method, called all-alive pruning (AAP), which seeks to effectively remove dead connections in a network. The proposed approach aims at enhancing the saliency-based pruning, and technically the approach searches for the dead neurons by inspecting their gradient flows.

Reason for score:
Overall, I recommend accepting this manuscript. Although the proposed solution (AAP) is simple, the experimental results on several different saliency-based pruning scenarios consistently demonstrate its effectiveness and versatility.

Pros:
- The motivation was effectively described and the problem was well-defined which made it easy to read.
- versatile and applicable, state-of-the-art

Cons/Questions:
- In Section 2, the authors seem to be differentiating between the terms "useless connections" and "dead connections". However, In 3.1, they make use of those two terms in a similar manner.
- Regarding the pruning steps introduced in Sec. 3.2 (specifically the "Repeat" step), I would be interested in getting more information about the convergence during iteration. What would be the most significant factor for the convergence?
- Authors repeatedly mention that one of the positive byproducts of AAP is that it is more compatible with modern memory architecture. Additional descriptions would be appreciated.
- In Section 4.1, the authors "conjecture that having the skip connections is advantageous when training after deleting the ResBlock". A bit difficult to grasp the purpose of this sentence, along with the previous sentences.

---

> ### Author Response · Authors · 2020-11-23
> **Response to R3**
>
> We sincerely appreciate your comments and efforts in reviewing our paper. Mostly, thanks for the positive comments about clear motivations (R3, R4), the effectiveness of the methods (R1, R2, R3, R4), and the good-written paper(R2, R3, R4). We revised our manuscript and added a study on the convergence of AAP and an iterative approach to obtain a sub-network with higher saliency scores in the Appendix.
>
> We responded to questions/concerns from the reviewer below:
>
> ---
>
> **Q1.** The terms' useless connection' and 'dead connections' in paper.
>
> **A1.** In the paper, we use the term dead connections to mean the useless connections that cannot contribute to the training process. However, as you mentioned, there may be some mixed-use of the terms 'useless connections' and 'dead connections,' especially for explaining the dynamic pruning.
>
> In this revision, we modify the description of the dynamic pruning in Section 2 to make it clear. Thank you for pointing out the confusion.
>
> ---
>
> **Q2.** The convergence of proposed AAP.
>
> **A2.** To avoid the dead connections alive at future iterations, we masked all dead connections to make them unable to revive at future iterations. This always guarantees the convergence of AAP by reducing the number of sub-network for consideration at each iteration. We add more details about the convergence in Appendix E, and also modify the Algorithm described in Appendix A to more correctly represent the processing of appearing dead connections.
>
> ---
>
> **Q3.** The positive byproducts of AAP in modern architecture.
>
> **A3.** Since AAP discovers the wasted input and output connections, some structural units such as neurons or filters that were being wasted can now be excluded with AAP. This allows us to directly reduce the size of the matrix similar to structured pruning even before gaining the advantage of the sparse matrix. For the final version, we will consider adding some experiments to validate how AAP can actually reduce the model size.
>
> ---
>
> **Q4.** The advantages of skip connection in pruning.
>
> **A4.** By the sentence 'We conjecture that having the skip connections is advantageous when training after deleting the ResBlock.', we tried to describe that the skip connection enabled us to perform well on high sparsity by transferring the output from previous layers even without the entire ResBlock. We modify the sentences you pointed out to clarify why we stated that ResBlock is more advantageous for pruning. Thank you for pointing out.

---

### Official Review · AnonReviewer4 · 2020-10-30
**Novelty is limited**

**Rating:** 4
**Confidence:** 4

**Review:**

This work proposes a novel pruning method, called all-alive pruning (AAP), which is a general technique to remove dead connections from pruned neural networks. AAP is broadly applicable to various saliency-based pruning methods and model architectures. AAP equipped with existing pruning methods consistently improves the accuracy of original methods on three benchmark datasets.

Strengths
1.	The motivation is very clear, and AAP is expected to improve existing pruning methods by removing dead connections especially with high compression ratios.
2.	The authors perform various experiments on three benchmark datasets changing experimental settings such as classifiers, base pruning methods, and compression ratios.
3.	The paper is written well. It is easy to understand, and the ideas and experiments are presented well.

Weaknesses
1.	There is no theoretical study of AAP. Algorithm 2 removes dead connections in a greedy manner by making the scores of dead weights as zero, even though they can be alive by revived connections at future iterations. Repeating Steps 1 & 2 of Section 3.2 until the unguaranteed convergence can make it remove important weights having high saliency scores at the worst scenario. Theoretical guarantees on the convergence of the algorithm and the scores of removed weights are needed.
2.	The experiments are not practical. The proposed approach works well only with high compression ratios which degrade the performance of original models. Considering that the objective of model compression is to maintain the original accuracy requiring less resources, such high compression ratios that decrease the accuracy more than 5 points do not seem practical in real-world scenarios.
3. The novelty is limited. Reviving dead connection is not a new idea.

---

> ### Author Response · Authors · 2020-11-23
> **Response to R4**
>
> We sincerely appreciate your comments and efforts in reviewing our paper. Mostly, thanks for the positive comments about clear motivations (R3, R4), the effectiveness of the methods (R1, R2, R3, R4), and the good-written paper(R2, R3, R4). We revised our manuscript and added a study on the convergence of AAP and an iterative approach to obtain a sub-network with higher saliency scores in the Appendix.
>
> We responded to questions/concerns from the reviewer below:
>
> ---
>
> **Q1.** The theoretical study of AAP.
>
> **A1-1.** To avoid the dead connections alive at future iterations, we masked all dead connections to make them unable to revive at future iterations. This always guarantees the convergence of AAP by reducing the number of sub-network for consideration at each iteration. Without this, as you are concerned, AAP may result in an unexpected sub-network that contains many connections with low saliency scores.
>
> We add more details about the convergence in Appendix E, and also modify the Algorithm described in Appendix A to more correctly represent the processing of appearing dead connections.
>
> **A1-2.** AAP can always converge, but it does not mean that it can guarantee the sub-network with the highest total saliency score. It is hard to find the optimal sub-network with the existing pruning methods. For example, the saliency-based pruning methods can not find the optimal sub-network if the lowest connection needs to be alive.
>
> However, a better scenario can be achieved by applying AAP earlier from the target pruning ratio. For example, we can obtain much higher saliency scores at 90% sparsity if we already remove the dead connections at 80% sparsity. We conducted some experiments with these settings in Appendix E. Please refer Appendix E for more details.
>
> ---
>
> **Q2.** The practicality of AAP in the real-world.
>
> **A2.** In this work, we mainly focus on the extremely high sparsity. Recently, it has been shown that the performance of the model can be maintained sufficiently in the wild sparsity of about 50~80%, so the number of studies aiming to improve the accuracy at higher sparsity are increasing. Such researches[1,2] expect that the better it works at high sparsity, the better the saliency criteria will be.
>
> The unstructured pruning in wild ratios also have some concerns: 1) Due to the nature of the unstructured pruning, it is difficult to obtain an advantage of a sparse matrix in a wild ratio. 2) As the size of recent models grows exponentially, there are still a huge number of parameters left after pruning.
>
> Therefore we pretend to show that the existing unstructured pruning can achieve higher performance at higher sparsity with the simple method, and we hope that this could be a new baseline of future studies.
>
> ---
>
> **Q3.** The novelty of reviving the dead connections.
>
> **A3.** We haven't revived the dead connections in the paper, but we believe that your concern is from the 'revive' term. There are several pieces of research that use the term 'reviving' for their methods. For example, the dynamic pruning iteratively revives connections while pruning unimportant connections. However, to our best knowledge, there are no attempts to revive trainable connections by removing the connections that cannot contribute to the regular training, which occurs after applying the existing pruning methods.
>
> ---
>
> [1] Ning Liu, Xiaolong Ma, Zhiyuan Xu, Yanzhi Wang, Jian Tang, and Jieping Ye. Autocompress: An automatic DNN structured pruning framework for ultra-high compression rates. In AAAI Conference on Artificial Intelligence, pp. 4876–4883, 2020.
>
> [2] Utku Evci, Trevor Gale, Jacob Menick, Pablo Samuel Castro, and Erich Elsen. Rigging the lottery: Making all tickets winners. CoRR, abs/1911.11134, 2019.

---

### Decision · Program_Chairs · 2021-01-07
**Final Decision**

**Decision:**

Reject

**Comment:**

This paper introduces All-Alive Pruning, an approach which checks for and removes the connections to and from units with zero gradient ("dead" units). The method is shown to improve performance of IMP at extreme (>128x) compression ratios on MNIST, CIFAR-10, and Tiny ImageNet. All reviewers felt that the problem the authors study -- how to identify and remove dead units -- is an interesting one.

However, there were concerns about the practical utility of the method, given that AAP only improves performance for extreme compression ratios in which performance is already substantially degraded relative to unpruned models. I share these concerns, which mute the practical impact of this work. There were also concerns about a lack of proper baseline comparisons to more simple approaches to removing dead units. As mentioned by R1, given that the problem the study is an interesting one, the paper could make up for the lack of practical utility by providing detailed analyses of the settings in which dead units emerge, differences among pruning approaches, etc., but analyses provided here are limited.

I would encourage the authors to explore these areas in a future revision of the paper, but recommend that the paper be rejected in its current form.